# IA-MARL: Imputation Assisted Multi-Agent Reinforcement Learning for Missing Training Data

## Abstract

Recently, multi-agent reinforcement learning (MARL) adopts the centralized training with decentralized execution (CTDE) framework that trains agents using the data from all agents at a centralized server while each agent takes an action from its observation. In the real world, however, the training data from some agents can be unavailable at the centralized server due to practical reasons including communication failures and security attacks (e.g., data modification), which can slow down training and harm performance. Therefore, we consider the *missing training data problem* in MARL, and then propose the imputation assisted multi-agent reinforcement learning (IA-MARL). IA-MARL consists of two steps: 1) *the imputation of missing training data*, which uses generative adversarial imputation networks (GAIN), and 2) *the mask-based update of the networks*, which trains each agent using the training data of corresponding agent, not missed over consecutive times. In the experimental results, we explore the effects of the data missing probability, the number of agents, and the number of pre-training episodes for GAIN on the performance of IA-MARL. We show IA-MARL outperforms a decentralized approach and even can achieve the performance of MARL without missing training data when sufficient imputation accuracy is supported. Our ablation study also shows that both the mask-based update and the imputation accuracy play important roles in achieving the high performance in IA-MARL.

## 1 Introduction

Reinforcement learning (RL) solves many challenging problems including the game playing (Mnih et al., 2015) and the robot control (Levine et al., 2016), which focus on the single-agent RL environment, modeled as the Markov decision process (Sutton and Barto, 2011). However, there exist many real-world problems that involve interaction among multiple agents such as multi-robot control (Hüttenrauch et al., 2019) and multiplayer games (Silver et al., 2017; Bard et al., 2020). Hence, the multi-agent reinforcement learning (MARL) that operates in multi-agent domain has been introduced and now becomes one of the most active and challenging RL research areas.

In MARL, the decentralized approach has been used to train each agent based on its trajectory (Tan, 1993). However, it often shows unstable and low performance due to non-stationary environment and partially observable information (Tan, 1993; Foerster et al., 2017) that inherits from the decentralization. Specifically, as the agents evolve their policies independently, the environment becomes non-stationary, which unstabilizes training at each agent. In addition, the agent may not observe the information of other agents, which causes low performance in the cooperative or competitive environment (Lowe et al., 2017).

Recently, the centralized training with decentralized execution (CTDE) framework has been introduced for MARL (Oliehoek et al., 2008; Foerster et al., 2018). This can alleviate the non-stationary environment and the partially observable information problems (Lowe et al., 2017) and encourages coordination among agents (Foerster et al., 2018). In the execution of CTDE, each agent takes an action based on its observation, while the training of the agents is performed at a centralized server after collecting observations, actions, and rewards of all agents. In existing works, those data from all agents are assumed to be available at the centralized server, which may not be always true in reality.

The data from distributed agents can be unavailable due to practical reasons including the communication failure, hardware limit, and security attacks (Lakshminarayan et al., 1999; Twala, 2009). For instance, in wireless sensor applications of MARL such as vehicle tracking (Liang et al., 2020) and environmental monitoring (Li et al., 2020), sensors (i.e., agents) transmit their sensed information to a receiver (i.e., centralized server) for training. The training data, transmitted from sensors, can be missed when the communication is unstable. In addition, even when the training data successfully arrives at the centralized server, certain data can be removed from the training dataset due to security attacks such as false data injection (Yan et al., 2016) and unauthorized data modification (Ferretti et al., 2014). As one can readily imagine, this missing training data can cause a serious problem in MARL as the training cannot be performed.

One possible solution on this *missing data problem* is to use only the training data that contains data from all agents without missing. However, in this case, the number of training data can dramatically decrease as the number of agents increases or the data missing happens more often, which delays the training. Another solution can be the use of imputation for replacing the missing training data in the MARL. However, the training data of this case can be different from the original data, which potentially degrades the performance. Therefore, as discussed above, the missing data problem should be carefully considered for bringing the MARL to the next level for wider range of applications. Despite of it, to the best of our knowledge, the missing data problem in MARL has not been taken into account in existing works.

In this paper, we propose an imputation assisted multi-agent reinforcement learning (IA-MARL) with considering the missing training data problem, where the training data of each agent, consists of observation, action, and reward, can be randomly missed with certain probability. The proposed IA-MARL consists of two steps: 1) the imputation of missing training data and 2) the mask-based update of the networks. Specifically, for *the imputation of the missing training data*, we use a generative adversarial imputation network (GAIN) to impute the data from all agents, and we form the data for the training of agents. We then perform *the mask-based update*, which trains the value function and the policy of each agent by selectively using the training data of the corresponding agent, not missed over the consecutive times. In the experimental results, we show the IA-MARL outperforms a decentralized approach and also can achieve the performance of MARL with all training data without missing. We then also show the performance of IA-MARL for different missing probabilities, the number of agents, and the number of pre-training episodes for GAIN. From the ablation study, we also verify the importance of the mask-based update as well as the imputation accuracy in multi-agent environments when the training data can be missed.

## 2 RELATED WORK

Independent Q-learning has been proposed as a decentralized approach to train each agent using its own data independently (Tan, 1993). The independent Q-learning is used for the tabular environment (Littman, 1994), and deep learning-based approaches are presented in Tampuu et al. (2017); Gupta et al. (2017). Independent Q-learning, however, suffers from the non-stationary environment and partially observable information problems.

CTDE is one of the solutions for those problems. In CTDE, the data from all agents are used for the centralized training, while the execution at each agent only requires its observation (Oliehoek et al., 2008). For instance, the centralized server trains the value function of each agent using observations, actions, and rewards of all agents, while each agent takes an action based on its observation (Lowe et al., 2017) .

Recently, MARL algorithms that adopt CTDE framework have been presented. For instance, Lowe et al. (2017) proposes the multi-agent deep deterministic policy gradient (MADDPG) that extends deep deterministic policy gradient (DDPG) for the continuous control of multiple agents. For the credit assignment problem, the counterfactual baseline (Foerster et al., 2018) and the value function factorization that determines the contribution of each agent are used (Sunehag et al., 2018; Rashid et al., 2018; Son et al., 2019). To improve the performance of MARL, the soft value function and the multi-head attention are used in Iqbal and Sha (2019), and the communication between agents that provides additional information to each agent is introduced in Foerster et al. (2016); Mordatch and Abbeel (2018) while the limited communication channel between agents during the execution

is considered in Kim et al. (2019) to address the real world communication channel constraints in MARL. However, none of the prior work considers the missing training data problem.

Imputation is the research area that replaces missing data, which has been used for many applications including medical data and image concealment (Rubin, 2004). For the imputation, some techniques such as multivariate imputation by chained equations (MICE) (Buuren and Groothuis-Oudshoorn, 2010), matrix completion (Mazumder et al., 2010), and MissForest (Stekhoven and Bühlmann, 2012) are proposed. However, when the imputation is used for the data that have large data space (e.g., the data obtained from agents in CTDE), the techniques with insufficient expressive power might result in low performance. For this case, some imputation techniques that have more expressiveness by adopting the deep neural networks can be more suitable such as the multiple imputation using denoising autoencoders (MIDA) (Gondara and Wang, 2018), the bidirectional recurrent imputation for time series (BRITS) (Cao et al., 2018), and the generative adversarial imputation network (GAIN) (Yoon et al., 2018).

## 3 BACKGROUND

We consider a decentralized partially observable Markov decision process, defined by a tuple $(S, A, P, r, \Omega, O, \gamma, n)$, where $S$, $A$, and $\Omega$ are set of states, actions, and observations, respectively. $r$ and $\gamma$ are the reward and the discount factor, respectively, and $n$ is the number of agents. We use $s \in S$, $a \in A$, and $o \in \Omega$ for a state, an action, and an observation, respectively. We use subscript $i$ for the corresponding agent and $t$ for the time, e.g., $o_{i,t}$ is the observation of agent $i$ at time $t$. We use bold symbols to denote observations, actions, and rewards of all agents, e.g., $\mathbf{a}_t = (a_{1,t}, \cdots, a_{n,t})$. Here, $P(s_{t+1}|s_t, \mathbf{a}_t)$ and $O(\mathbf{o}_t|s_t)$ are the transition probability and the conditional observation probability, respectively.

### 3.1 DDPG AND MADDPG

The objective of the agent in the environment is to maximize the cumulative reward $R_t = \sum_{t'=t}^{T} \gamma^{t'-t} r_{t'}$. For this, we use the actor-critic method. The expected cumulative reward for given action and state is $Q(s_t, a_t) = \mathbb{E}[R_t|s = s_t, a = a_t]$, which is called as an action-value function or value function. Using the Bellman equation, the value function can be rewritten as $Q(s_t, a_t) = \mathbb{E}_{r_t, s_{t+1}, a_{t+1}}[r_t + \gamma Q(s_{t+1}, a_{t+1})]$. When the parameter $\theta$ is used for the value function approximation, the value function $Q$ can be learned by minimizing the loss $\mathcal{L}(\theta)$, given as

$$\mathcal{L}(\theta) = \mathbb{E}\left[(Q_\theta(s_t, a_t|\theta) - y)^2\right], \ y = r_t + \gamma Q_\theta(s_{t+1}, a_{t+1}). \tag{1}$$

DDPG is the widely-used choices for the policy update. The policy parameterized by $\phi$ takes state $s$ as an input and outputs deterministic action $a = \mu_\phi(s)$ in DDPG, where the gradient of $\phi$ is given as

$$\nabla_\phi J(\phi) = \mathbb{E}[\nabla_\phi \mu_\phi(a_t|s_t)\nabla_{a_t} Q_\theta(s_t, a_t)|_{a_t=\mu_\phi(s_t)}]. \tag{2}$$

MADDPG is an algorithm that uses DDPG in the CTDE framework (Lowe et al., 2017). The value function in MADDPG takes observations and actions of all agents as an input. Meanwhile, the policy of each agent takes its observation as an input since the agent can access to its observation only. In CTDE, the loss for the value function and the gradient for the policy are given as

$$\mathcal{L}(\theta_i) = \mathbb{E}[(Q_{\theta_i}(\mathbf{o}_t, \mathbf{a}_t) - y)^2], \ y = r_{i,t} + \gamma Q_{\theta_i}(\mathbf{o}_{t+1}, \mathbf{a}_{t+1}), \tag{3}$$

$$\nabla_{\phi_i} J(\phi_i) = \mathbb{E}[\nabla_{\phi_i} \mu_{\phi_i}(a_{i,t}|o_{i,t})\nabla_{a_{i,t}} Q_{\theta_i}(\mathbf{o}_t, \mathbf{a}_t)|_{a_{i,t}=\mu_{\phi_i}(o_{i,t})}]. \tag{4}$$

Here, each agent has a different value function and a policy, parameterized respectively as $\theta_i$ and $\phi_i$.

### 3.2 IMPUTATION

In statistics, imputation is used to replace missing data with substituted values. In order to denote the missingness of the data, the mask $M$ that has the same dimension as the data $X$ is used: the $k$-th component of the mask $m_k = 0$, when the $k$-th data $x_k$ is missed, and $m_k = 1$, when $x_k$ is not missed. We denote the obtained data as $M \odot X$, where $\odot$ is elementwise multiplication. In the imputation, the completed data $\hat{X}$ is given as

$$\hat{X} = M \odot X + (1 - M) \odot \bar{X}, \tag{5}$$

where $\bar{X}$ is the imputed data using the imputation algorithm including GAIN. For the loss of the neural network that imputes data, the mean squared error is used, given as

$$\mathcal{L}_M(\bar{X}, X) = \sum_k m_k(\bar{x}_k - x_k)^2. \tag{6}$$

Here, $\bar{x}_k$ is $k$-th component of the imputed data. Note the loss can be estimated when $m_k = 1$ since $x_k$ can be used only when it is not missed.

## 4 METHODS

In this section, after introducing the missing training data problem in MARL and describing the imputation method, we propose IA-MARL.

### 4.1 MISSING TRAINING DATA PROBLEM AND IMPUTATION METHOD

We consider the MARL environment where the training data of each agent is randomly missed, called the missing training data problem. Specifically, the training data from agent $i$ at time $t$, $\tau_{i,t} = (o_{i,t}, a_{i,t}, r_{i,t})$, is missed with the probability $p_{m_{i,t}} = \mathbb{P}(m_{i,t} = 0) < 1$, [1] as well. where $m_{i,t} \in \{0, 1\}$ denotes whether the data of agent $i$ at time $t$ is missed ($m_{i,t} = 0$) or not ($m_{i,t} = 1$).

In the presence of missing data, to obtain completed data of all agents at time $t$ for training, we first set the data for the imputation and the mask, respectively, as

$$X_t = (\boldsymbol{\tau}_{t-1}, \boldsymbol{\tau}_t, \boldsymbol{\tau}_{t+1}), \ M_t = (\mathbf{m}_{t-1}, \mathbf{m}_t, \mathbf{m}_{t+1}), \tag{7}$$

where $\boldsymbol{\tau}_t = (\tau_{1,t}, \cdots, \tau_{n,t})$ and $\mathbf{m}_t = (m_{1,t}J_{1,|\tau_{1,t}|}, \cdots, m_{n,t}J_{1,|\tau_{n,t}|})$. Here, $|\cdot|$ is a cardinality of set $\cdot$ and $J_{1,|\tau|}$ is an all-ones vector with length $|\tau|$. Note that for the accurate imputation of $\tau_{i,t}$, the temporal data correlation as well as the data correlation across different agents should be used.[2]

For the imputation, we use GAIN (Yoon et al., 2018) as it has sufficient expressiveness to impute multi-agent data, does not require original data for the training, and also has state-of-art performance. The generator $G$ in GAIN takes the obtained data, the mask matrix, and the random matrix as inputs and outputs the imputed data. The imputed data and the completed data can be presented, respectively, as

$$\bar{X}_t = G(M_t \odot X_t, M_t, (1 - M_t) \odot Z_t), \tag{8}$$

$$\hat{X}_t = M_t \odot X_t + (1 - M_t) \odot \bar{X}_t, \tag{9}$$

where $Z_t$ is a random matrix that has the same dimension as $X_t$. In equation 9, the completed data can be represented as $\hat{X}_t = (\hat{\boldsymbol{\tau}}_{t-1}, \hat{\boldsymbol{\tau}}_t, \hat{\boldsymbol{\tau}}_{t+1})$. The discriminator $D$ in GAIN takes $\hat{X}_t$ and the hint matrix $H_t$ as inputs and outputs the probability of being masked for the completed data as

$$\hat{M}_t = D(\hat{X}_t, H_t), \tag{10}$$
$$H_t = B_t \odot M_t + 0.5(1 - B_t).$$

In equation 10, $B_t = (b_{1,t-1}, \cdots, b_{n,t+1})$ is a random sequence of 0 and 1, where $p_h = \mathbb{P}(b_{i,t} = 1)$ is the hint probability.

---

**Algorithm 1** GAIN Training and Imputation

**Require:** Dataset $\mathcal{D}_{\text{GAIN}}$ contains $X_t$ and $M_t$
1: Generate random matrix $Z_t$ and hint matrix $H_t$
2: Generate completed data $\hat{X}_t$ using equation 9
3: Discriminate completed data using equation 10
4: Estimate $\mathcal{L}_D$ using equation 11 and update discriminator
5: Generate random matrix $Z_t$ and hint matrix $H_t$
6: Generate completed data $\hat{X}_t$ using equation 9
7: Discriminate completed data using equation 10
8: Estimate $\mathcal{L}_G$ using equation 12 and update generator
9: Return $\hat{X}_t$

---

[1]We do not consider the case of missing all training data of all agents and time since the imputation cannot proceed. We also note that the proposed IA-MARL can solve the problems of missing a certain part of $\tau_{i,t}$ (e.g., missing $o_{i,t}$ or $a_{i,t}$)

[2]One may use $X_t = (\boldsymbol{\tau}_{i,t-t'}, \cdots, \boldsymbol{\tau}_{i,t+t'})$, $t' > 1$. As $t'$ increases, $X_t$ contains more information which may increase the imputation accuracy. At the same time, as $t'$ increases, the input and output dimensions of GAIN increase which may decrease the imputation accuracy. Due to the ambiguous effect of $t'$ on the imputation accuracy, we left it as a design choice and we use $t' = 1$.

The role of the hint matrix is to make the discriminator focus on the component with $b_{i,t} = 0$. The objective of the discriminator $D$ is to maximize the probability of estimating the mask matrix correctly, while the objective of the generator $G$ is to minimize the both accuracy of the discriminator and the loss of the imputation. Therefore, the losses of $D$ and $G$ are, respectively, given by

$$\mathcal{L}_D = \mathbb{E}_{(X_t, M_t) \in \mathcal{D}_{\text{GAIN}}} \left[ - \sum_{i,t:b_{i,t}=0} (m_{i,t} \log(\hat{m}_{i,t}) + (1 - m_{i,t}) \log(1 - \hat{m}_{i,t})) \right], \qquad (11)$$

$$\mathcal{L}_G = \mathbb{E}_{(X_t, M_t) \in \mathcal{D}_{\text{GAIN}}} \left[ - \sum_{i,t:b_{i,t}=0} (1 - m_{i,t}) \log(\hat{m}_{i,t}) + \alpha_G \mathcal{L}_M(\bar{X}_t, X_t) \right], \qquad (12)$$

where $\mathcal{D}_{\text{GAIN}}$ is the dataset that contains obtained data and its mask matrix, and $\hat{m}_{i,t} \in [0, 1]$ is the probability of being masked, i.e., the $i$-th component of $\hat{M}_t$ in equation 10. In equation 12, $\mathcal{L}_M(\bar{X}_t, X_t)$ is the difference between the imputed data and the obtained data as in equation 6. The parameter $\alpha_G$ controls the importance of the imputation loss, which is a hyperparameter. The training and the imputation of GAIN are given in Algorithm 1.

## 4.2  IA-MARL: IMPUTATION ASSISTED MULTI-AGENT REINFORCEMENT LEARNING

In this subsection, we propose IA-MARL, which uses the completed data obtained from GAIN and the mask-based update that stabilizes the training in MARL for the missing training data problem. For IA-MARL, we use MADDPG, but other actor-critic RL methods including policy gradient are also applicable for IA-MARL. Note that with some modification on the value function update, IA-MARL can also be used for the value-based MARL algorithms including VDN and QMIX.

In IA-MARL, the value function and the policy of agent $i$ can be updated using the following loss and gradient.

$$\mathcal{L}(\theta_i) = \mathbb{E}_{\hat{\tau}_t \sim \hat{\mathcal{D}}_i} \left[ (Q_{\theta_i}(\hat{\mathbf{o}}_t, \hat{\mathbf{a}}_t) - y)^2 \right],$$
$$y = \hat{r}_{i,t} + \gamma Q_{\theta'_i}(\hat{\mathbf{o}}_{t+1}, \hat{\mathbf{a}}_{t+1}), \qquad (13)$$
$$\nabla_{\phi_i} J(\phi_i) = \mathbb{E}_{\hat{\tau}_t \sim \hat{\mathcal{D}}_i} [\nabla_{\phi_i} \mu_{\phi_i}(\hat{a}_{i,t}|\hat{o}_{i,t}) \qquad (14)$$
$$\nabla_{\hat{a}_{i,t}} Q_{\theta_i}(\hat{\mathbf{o}}_t, \hat{\mathbf{a}}_t)|_{\hat{a}_{i,t} = \mu_{\phi'_i}(\hat{o}_{i,t})}],$$

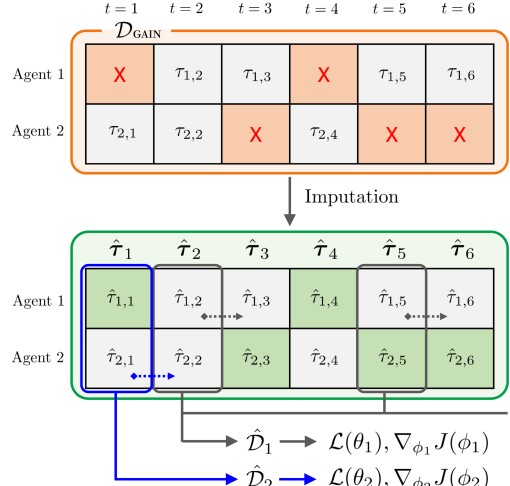

Figure 1: An example of the mask-based update in IA-MARL when $n = 2$.

where $\theta'_i$ and $\phi'_i$ indicate the target network parameters periodically updated as in (Hasselt et al., 2016). Even though we use imputation, the completed data may have different value with the original data, which harms the performance. To mitigate this, we propose the mask-based update that trains the value function and the policy using $\hat{\mathcal{D}}_i$, which is the set of the completed data for the training of agent $i$, given by

$$\hat{\mathcal{D}}_i = \{\hat{\tau}_t \mid m_{i,t} m_{i,t+1} = 1, t \in \{1, \cdots, T_{\max} - 1\}\} \cup \{\hat{\tau}_{T_{\max}} \mid m_{i,T_{\max}} = 1\}, \qquad (15)$$

where $T_{\max}$ is an episode length. Note that $\hat{\mathcal{D}}_i$ in equation 15 contains data collected across episodes, and $\hat{\tau}_{T_{\max}}$ is included in $\hat{\mathcal{D}}_i$ when $m_{i,T_{\max}} = 1$ to update $\theta_i$ and $\phi_i$ at time $T_{\max}$. The mask-based update means, in the training of agent $i$, the completed data at time $t$ is used only when the data of agent $i$ is not missed over two consecutive times, $t$ and $t + 1$. An example of $\hat{\mathcal{D}}_i$ is presented in Fig. 1. As shown in Fig. 1, only the completed data at time 2 and 5 are included in $\hat{\mathcal{D}}_1$, and the completed data at time 1 is included in $\hat{\mathcal{D}}_2$.

In IA-MARL, the imputation and the mask-based update are used to make the policy and the value function (i.e., $\mu_{\phi_i}(\hat{a}_{i,t}|\hat{o}_{i,t})$ and $Q_{\theta_i}(\hat{\mathbf{o}}_t, \hat{\mathbf{a}}_t)$) similar to the ones without data missing (i.e., $\mu_{\phi_i}(a_{i,t}|o_{i,t})$ and $Q_{\theta_i}(\mathbf{o}_t, \mathbf{a}_t)$), respectively. In case of the policy, as $m_{i,t} = 1$ (i.e., the data of agent $i$

at time $t$ exists) by the mask-based update, we can have $\mu_{\phi_i}(\hat{a}_{i,t}|\hat{o}_{i,t}) = \mu_{\phi_i}(a_{i,t}|o_{i,t})$. In case of the value function, since $m_{i,t}m_{i,t+1} = 1$ by the mask-based update, the value function can be updated when $(o_{i,t}, a_{i,t}, r_{i,t}, o_{i,t+1}, a_{i,t+1})$ exists that mainly affect the update of $Q_{\theta_i}(\hat{\mathbf{o}}_t, \hat{\mathbf{a}}_t)$. Hence, if the imputation on the missing data of other agents is accurate, $Q_{\theta_i}(\hat{\mathbf{o}}_t, \hat{\mathbf{a}}_t)$ can be similar to $Q_{\theta_i}(\mathbf{o}_t, \mathbf{a}_t)$. We also show the importance of the mask-based update and accurate imputation in Section 5.3.

The training algorithm for IA-MARL is provided in Algorithm 2. In IA-MARL, GAIN is periodically trained and outputs completed data using Algorithm 1, where the dataset for GAIN, $\mathcal{D}_{\text{GAIN}}$, is initialized to prevent the overfitting of GAIN to the trajectory of outdated policies. We pre-train GAIN through $N_{\text{pre}}$ episodes for better imputation accuracy. After $N_{\text{pre}}$ episodes, we initialize the parameters of all agents, i.e., $\phi_i, \phi_i', \theta_i, \theta_i'$, and $\hat{\mathcal{D}}_i, \forall i$, to prevent the agent from training with inaccurate data. The components in Algorithm 2 can be modified according to the MARL algorithm, where we use MADDPG in this work. Therefore, the training procedure follows that in Lowe et al. (2017), e.g., the action is selected as $a_i = \mu_{\phi_i}(o_i) + \mathcal{N}$, where $\mathcal{N}$ is noise for exploration.

---

**Algorithm 2** IA-MARL

1: Initialize $\mathcal{D}_{\text{GAIN}}$ and GAIN $G, D$
2: Initialize $\phi_i, \phi_i', \theta_i, \theta_i', \hat{\mathcal{D}}_i \; \forall i$
3: **for** episode $= 1$ to $N$ **do**
4:     **if** episode $= N_{\text{pre}}$ **then**
5:         Initialize $\mathcal{D}_{\text{GAIN}}, \hat{\mathcal{D}}_i, \phi_i, \phi_i', \theta_i, \theta_i' \; \forall i$
6:     **end if**
7:     Initialize environment and receive $\mathbf{o}$
8:     **for** $t = 1$ to $T_{\max}$ **do**
9:         For each agent $i$, select action $a_i$
10:         Store $\mathbf{o}, \mathbf{a}, \mathbf{r}, \mathbf{m}$ in $\mathcal{D}_{\text{GAIN}}$
11:         **for all** agent $i$ **do**
12:             Train $\phi_i$ and $\theta_i$ using equation 13 and equation 14
13:             Update target network $\phi_i'$ and $\theta_i'$
14:         **end for**
15:     **end for**
16:     Train GAIN using $\mathcal{D}_{\text{GAIN}}$ and update $\hat{\mathcal{D}}_i$, then initialize $\mathcal{D}_{\text{GAIN}}$
17: **end for**

---

In IA-MARL, the number of training data is smaller than the number of obtained data due to the mask-based update. When the missing probability of agent $i$ over time is equal, i.e., $p_{m_{i,t}} = p_{m_i}, \forall t$, and the number of obtained data is $N_{\mathcal{D}}$, the number of completed data for the training of agent $i$ is given as

$$\mathbb{E}\big[|\hat{\mathcal{D}}_i|\big] = N_{\mathcal{D}}(1 - p_{m_i})^2, \tag{16}$$

since the condition in equation 15 is satisfied when the data of agent $i$ exists during the consecutive times. Note that $\mathbb{E}\big[|\hat{\mathcal{D}}_i|\big]$ decreases with $p_{m_i}$.

When the imputation is not applied, the centralized server can only use the data which is not missed from all agents over consecutive times, i.e., $\mathcal{D}_{\mathcal{P}} = \{\boldsymbol{\tau}_t \mid m_{i,t}m_{i,t+1} = 1, \forall i, t \in \{1, \cdots, T_{\max} - 1\}\} \cup \{\boldsymbol{\tau}_{T_{\max}} \mid m_{i,T_{\max}}, \forall i\}$. In this case, the imputation is not required for the training. Hence, for given $N_{\mathcal{D}}$, the number of the training data is given as

$$\mathbb{E}\big[|\mathcal{D}_{\mathcal{P}}|\big] = N_{\mathcal{D}} \prod_{i=1}^{n} (1 - p_{m_i})^2. \tag{17}$$

Therefore, the number of training data without imputation, $\mathbb{E}\big[|\mathcal{D}_{\mathcal{P}}|\big]$, decreases exponentially with the number of agents $n$ while the number of training data for IA-MARL, $\mathbb{E}\big[|\hat{\mathcal{D}}_i|\big]$, is not affected by $n$. We show the performance of MADDPG without imputation in Appendix B.

## 5 EXPERIMENTAL RESULTS

In this section, we provide MARL environment and hyperparameters, and then show the performance of IA-MARL with different missing probabilities, the number of agents, and the number of pre-training episodes for GAIN. Furthermore, we provide the ablation study which shows the importance of the mask-based update and accurate imputation in IA-MARL.[3]

---

[3]The code is in the supplementary materials and will be published after the review.

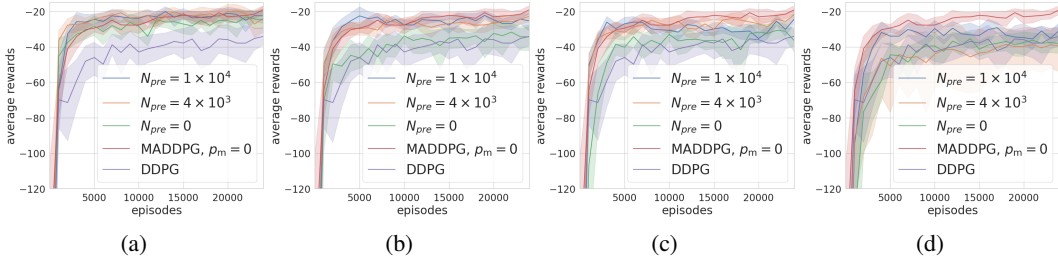

Figure 2: Speaker-Listener environment with a pair of speaker-listener when different training data missing probability $p_{\mathrm{m}}$ and number of pre-training episodes $N_{\mathrm{pre}}$ are considered. The training data missing probability is (a) $p_{\mathrm{m}} = 0.1$, (b) $p_{\mathrm{m}} = 0.2$, (c) $p_{\mathrm{m}} = 0.3$, and (d) $p_{\mathrm{m}} = 0.4$. Figure best viewed in color.

## 5.1 ENVIRONMENT AND HYPERPARAMETERS

Among widely-used MARL benchmark environments including the multi-agent particle environments (MPE) (Lowe et al., 2017; Mordatch and Abbeel, 2018) and the starcraft multi-agent challenge (SMAC) (Samvelyan et al., 2019), we use and modify the MPE that contains mixed cooperative and competitive environments. Note that SMAC is not used in experiments since MADDPG, adopted in IA-MARL, generally does not perform well in SMAC (Papoudakis et al., 2021).

The speaker-listener environments are composed of several pairs of agents, where each pair has a speaker and a listener, and landmarks have unique colors. In the environment, the speaker observes the target color and sends a message to the listener. The listener then moves after observing the locations of landmarks and the message. Each speaker-listener pair gets shared rewards, which is determined by the distance between the listener and the target landmark. Note that the speaker should learn which message to send, and the listener, who does not know the target landmark, should learn the message. We set the number of landmarks as $2 + n/2$ so that the number of landmarks increases with the number of pairs (i.e., $n/2$).

The tag environments are composed of one prey and multiple predators, where the number of predators is set as 3, 4, and 5 (which means the number of agents are $n = 4, 5$, and 6, respectively). In the environment, the objective of the prey is to run away from the predators, and the objective of the predator is to catch the prey. Each time when the prey collides with the predator, predators get a shared reward while the prey is penalized.

We evaluate the performance of the algorithm across 10 runs with different random seeds and the shade in graphs represents the confident interval. We use MADDPG for IA-MARL. As baselines, we use MADDPG and DDPG without missing training data (i.e., $p_{\mathrm{m}} = 0$). We use the same hyperparameters in Lowe et al. (2017), except for the training frequency of the network parameters in IA-MARL. Specifically, while MADDPG updates every 100 samples and collects 1024 samples before making an initial update, IA-MARL updates every $100/(1 - p_{\mathrm{m}})^2$ samples and collects $1024/(1 - p_{\mathrm{m}})^2$ samples before making an initial update. Accordingly, the target network parameters are less frequently updated in IA-MARL. In the speaker-listener environment, we evaluate the performance of IA-MARL against MADDPG speaker and listener. In the tag environment, we evaluate IA-MARL against MADDPG predators and DDPG prey.

## 5.2 PERFORMANCE OF IA-MARL

Figure 2 shows the average rewards as a function of the episodes with different number of pre-training episode and training data missing probability in the speaker-listener environment with one speaker-listener pair. In Figs. 2a-2d, $p_{\mathrm{m}} = 0.1, 0.2, 0.3, 0.4$ is used, respectively. We observe that the performance of IA-MARL outperforms the decentralized approach, i.e., DDPG, and also can achieve the performance of MARL without missing training data, i.e., MADDPG with $p_{\mathrm{m}} = 0$, when GAIN is pre-trained sufficiently. Specifically, IA-MARL achieves the performance of MADDPG with any $N_{\mathrm{pre}}$ for $p_{\mathrm{m}} = 0.1$ and $N_{\mathrm{pre}} \geq 4 \times 10^3$ for $p_{\mathrm{m}} = 0.2$ and 0.3. When $p_{\mathrm{m}} = 0.4$, IA-MARL could not achieve the performance of MADDPG, but it is expected to achieve the similar performance of MADDPG with larger $N_{\mathrm{pre}}$. Note that as $p_{\mathrm{m}}$ increases, mainly due to the smaller number of

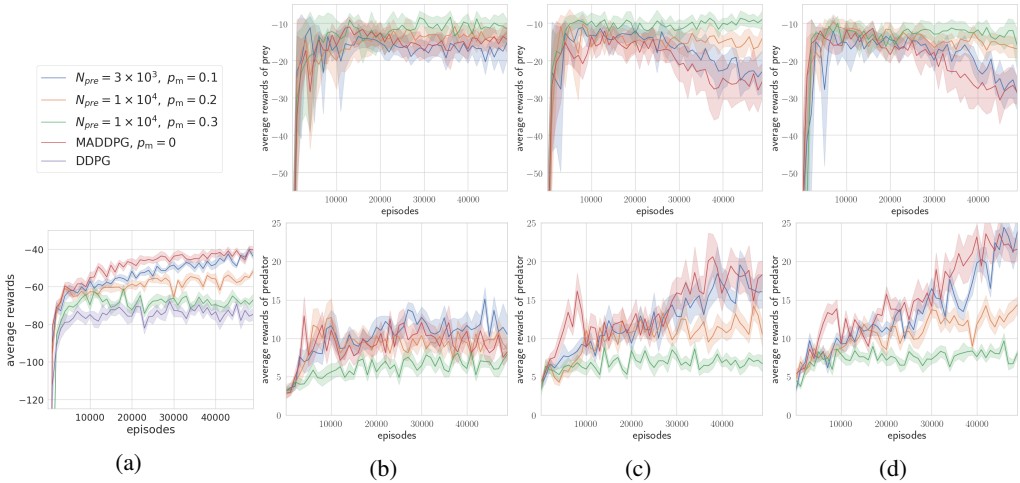

(a)  (b)  (c)  (d)

Figure 3: (a) Speaker-Listener environment with two pairs of agents. (b)-(d) Tag environment, where the upper graphs show the average rewards of the prey, and the lower graphs show the average rewards of the predator. The number of predators is 3 in (b), 4 in (c), and 5 in (d).

training data as shown in equation 16, IA-MARL slowly achieves the performance of MADDPG. Furthermore, as $p_{\mathrm{m}}$ increases, more pre-training episodes are required since GAIN is less trained due to the larger value of imputation loss. When $N_{\mathrm{pre}}$ is small, large imputation error in completed data causes IA-MARL to hardly achieve the performance of MADDPG.

Figure 3 shows the average rewards as a function of the episodes with different pre-training episode and missing probability. In Fig. 3a, the average rewards are shown for two pairs of speaker-listener. In Figs. 3b-3d, we show the average rewards when the number of predators is 3, 4, and 5, respectively, where the upper graphs show the rewards of the prey and the lower graphs show those of predators. We observe IA-MARL can achieve the performance of MADDPG, and more pre-training and training episodes are required as $p_{\mathrm{m}}$ increases. In Fig. 3a, when $p_{\mathrm{m}} = 0.1$ and $0.2$, IA-MARL achieves the performance of MADDPG slowly. Furthermore, Figs. 3b-3d also show IA-MARL achieves the performance of MADDPG when $p_{\mathrm{m}} = 0.1$, where the rewards of predators steadily increase, and the rewards of the prey decrease. However, as $p_{\mathrm{m}}$ increases, IA-MARL cannot achieve the performance of MADDPG due to large imputation error. For instance, in Fig. 3a, when $p_{\mathrm{m}} = 0.3$, the performance of IA-MARL is in between the performance of MADDPG and DDPG. Similarly, in Figs. 3c and 3d, when $p_{\mathrm{m}} = 0.2$ and $0.3$, IA-MARL cannot achieve the performance of MADDPG.

We also observe as the number of agents increases, more $N_{\mathrm{pre}}$ is required to make IA-MARL achieve the performance of MADDPG. In the speaker-listener environment with $p_{\mathrm{m}} = 0.3$, $N_{\mathrm{pre}} = 4 \times 10^3$ is required for $n = 2$ (see Figure 2a), while $N_{\mathrm{pre}} > 10^4$ is required for $n = 4$ (see Figure 3a). Similarly, in the tag environment with $p_{\mathrm{m}} = 0.2$, $N_{\mathrm{pre}} = 10^4$ is required for $n = 4$ (see Figure 3b), while $N_{\mathrm{pre}} > 10^4$ is required for $n = 5$ and $n = 6$ (see Figures 3c and 3d). The main reasons to require more $N_{\mathrm{pre}}$ for the larger number of agents are 1) environment with more agents becomes more complex and 2) the generator and the discriminator in GAIN require more training data as the input and output spaces increase with the number of agents.

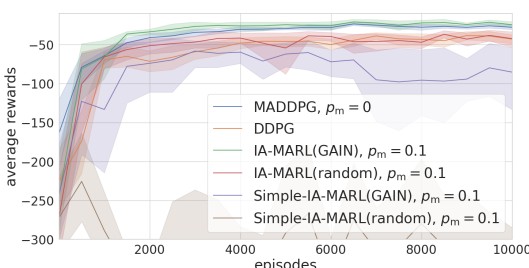

Figure 4: Speaker-Listener environment with a pair of agents. The average rewards with/without the mask-based update and with GAIN/random imputation are shown. We use $N_{\mathrm{pre}} = 10^4$ for GAIN.

## 5.3 ABLATION STUDY AND LIMITATION

We compare the performance of IA-MARL with and without the mask-based update and with and without GAIN imputation. For simplicity, we call IA-MARL without the mask-based update as simple-IA-MARL. In simple-IA-MARL, agent $i$ is trained using the completed data $\hat{\mathcal{D}} = \{\hat{\boldsymbol{\tau}}_t \mid t \in \{1, \cdots, T_{\max} - 1\}\}$, which is different from IA-MARL that trains agent $i$ using data $\hat{\mathcal{D}}_i$. When GAIN is not used, we use random imputation replacing missing training data with uniform random variables, i.e., $\hat{o}_{i,t} \sim \mathcal{U}(\min_{o_{i,t} \in \Omega_i}(o_{i,t}), \max_{o_{i,t} \in \Omega_i}(o_{i,t}))$, where $\Omega_i$ is the set of all $o_{i,t}$. Note the random imputation can be regarded as a simple imputation method that has low imputation accuracy. We show the performance of IA-MARL and Simple-IA-MARL with different imputation method and missing probability in Appendix B.

Figure 4 shows the ablation study of IA-MARL when $p_{\mathrm{m}} = 0.1$ and $N_{\mathrm{pre}} = 10^4$. Firstly, we observe the average rewards of simple-IA-MARL are even lower than that of DDPG for all imputation methods. The low performance of simple-IA-MARL is due to the effect of imputation error on the gradient estimation, which is alleviated in IA-MARL by the mask-based update. We also observe that the average reward of IA-MARL with random imputation is similar or lower than that of DDPG. Since the accuracy of random imputation is low, both the MARL with/without the mask-based update cannot achieve the performance of MARL without missing data. Similarly, when the number of pre-training episodes for GAIN, $N_{\mathrm{pre}}$, is small, the imputation accuracy is low, so the performance of IA-MARL is degraded as shown in Figs. 2-3. Therefore, when the imputation is not sufficiently accurate, the performance of IA-MARL can be low.

## 6 CONCLUSION

We propose IA-MARL for the training of agents in the presence of missing training data. The key idea is to use the imputation for replacing the missing data and the mask-based update that selectively uses the training data for each agent. In IA-MARL, we use GAIN for the imputation and MADDPG for MARL algorithm. In the experimental results, we verify the performance of IA-MARL for different training data missing probabilities, the number of agents, and the number of pre-training episodes for GAIN. We show that IA-MARL with missing training data achieves comparable performance with MADDPG without missing training data, when GAIN is pre-trained sufficiently. Through the ablation study, we also show the importance of the mask-based update and the imputation accuracy in IA-MARL for achieving high performance.

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

## A   HYPERPARAMETERS

We evaluate the performance of the algorithm across 10 runs with different random seeds. We use grid search to fine-tune the learning rate and the number of hidden nodes of MADDPG and DDPG. For GAIN, we use the generator and the discriminator consisting of 2 hidden layers with 128 nodes, where ReLU activation is used except for the last layers that use sigmoid. We use Adam optimizer with a learning rate of 0.001 for the training of GAIN. The hint probability $p_h = 0.5$ and $\alpha_G = 100$ are used, and GAIN is trained every 5 episodes. The replay buffer for GAIN is initialized after each training.

We use MADDPG for IA-MARL. Each agent has the value function and the policy consisting of 2 hidden layers with 64 nodes. We use ReLU for activations except for the last layer of value function that does not use activation and the last layer of policy that uses tanh activation. We use Adam optimizer for all agents with the learning rate of 0.01 and use target networks with $\tau = 0.01$. The length of the replay buffer is $10^6$. MADDPG updates every 100 samples and collects 1024 samples before making an initial update. Since the mask-based update makes the number of training data for IA-MARL smaller, we update IA-MARL every $100/(1 - p_{\mathrm{m}})^2$ samples and collects $1024/(1 - p_{\mathrm{m}})^2$ samples before making an initial update. Accordingly, the target network parameters are less frequently updated in IA-MARL.

## B   ADDITIONAL EXPERIMENTAL RESULTS

In this Appendix, to show the effectiveness of IA-MARL in terms of the number of training data discussed in Section 4.2, we show the training curves of MADDPG with missing training data and compare them with IA-MARL. Then we show the performance IA-MARL with different imputation methods. Furthermore, we evaluate the computational time of IA-MARL.

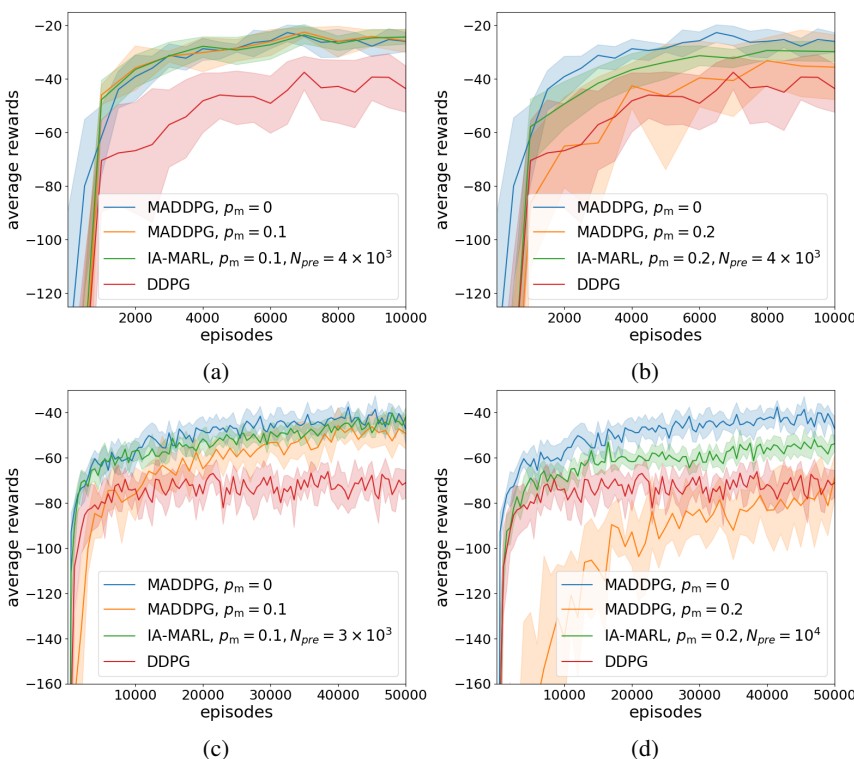

Figure 5: (a)-(b) Speaker-Listener environment with a pair of agents. (c)-(d) Speaker-Listener environment with two pairs of agents.

Figure 5 shows the performance of MADDPG with different $p_{\mathrm{m}}$ in Speaker-Listener environment with $n = 2$ and $n = 4$. We update MADDPG every $N_{\mathcal{D}} = 100/\{\prod_{i=1}^{n}(1 - p_{\mathrm{m}_i})^2\}$ steps since the number of data for the training decreases due to the data missing. When the training data is missed, except for Fig. 5a, MADDPG has lower performance than IA-MARL due to the small number of training data $\mathbb{E}\big[\big|\mathcal{D}_{\mathcal{P}}\big|\big]$ as in equation 17. Note that as the number of agents becomes larger, MADDPG is more slowly trained, while the number of training data of IA-MARL is not affected by the number of agents.

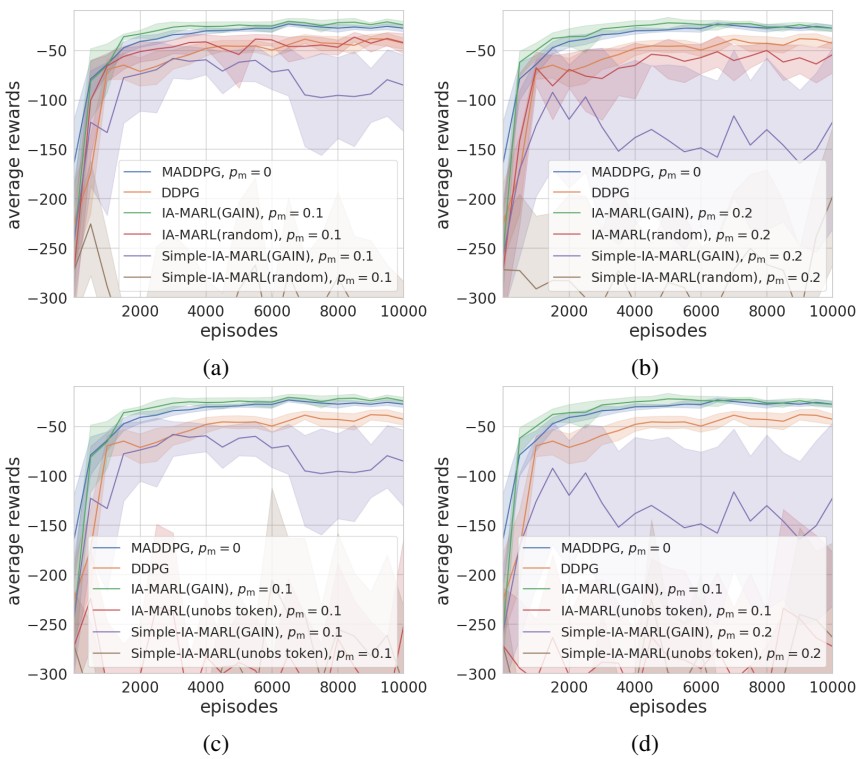

Figure 6: IA-MARL and Simple-IA-MARL with (a)-(b) random imputation and (c)-(d) unobserved token for the missing training data.

Figure 6 shows the performance of IA-MARL and Simple-IA-MARL with different imputation methods. In Figs. 6a and 6b, we replace the missing training data with uniform random variables, i.e., $\hat{o}_{i,t} \sim \mathcal{U}(\min_{o_{i,t} \in \Omega_i}(o_{i,t}), \max_{o_{i,t} \in \Omega_i}(o_{i,t}))$, where $\Omega_i$ is the set of all $o_{i,t}$. In Figs. 6c and 6d, we replace the missing training data with the unobserved token $-1$, i.e., $\hat{o}_{i,t} = -1$. Note that for the both IA-MARL and Simple-IA-MARL, the GAIN shows the better performance than the random imputation and the use of unobserved token for the missing training data.

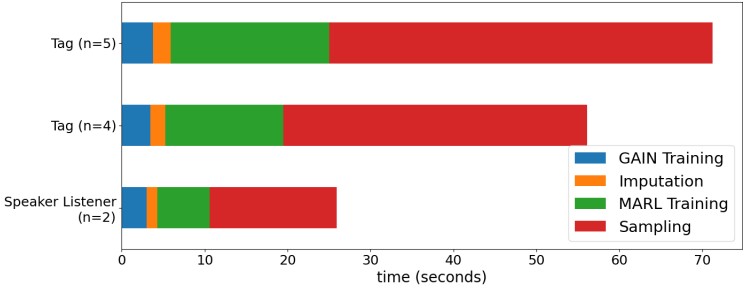

Figure 7: Computation time of IA-MARL across 1000 episodes.

Figure 7 shows the computation time of IA-MARL across 1000 episodes when the numbers of agents are 2, 4, and 5, respectively. We use Intel i7-9700K CPU and NVIDIA GeForce RTX 2080Ti GPU

for the evaluation. The time for sampling and MARL training takes a major portion compared to the time for the imputation and GAIN training. Note that as the number of agents increases, the time for sampling and MARL training linearly increases since each agent has the value function and the policy that should be trained. On the other hand, GAIN has two networks, i.e., the generator and the discriminator. Therefore, the time for imputation and GAIN training slowly increases with the number of agents compared with the time for sampling and MARL training.

