# OpenReview forum: "IA-MARL: Imputation Assisted Multi-Agent Reinforcement Learning for Missing Training Data"
_ICLR.cc/2022/Conference — ICLR 2022 Submitted_

### Official Review · Reviewer_vfgR · 2021-10-27

**Correctness:** 2
**Technical Novelty And Significance:** 2
**Empirical Novelty And Significance:** 2
**Recommendation:** 1
**Confidence:** 4

**Main Review:**

### Writing
The submission is poorly written, to say the least. I've quoted a couple of sentences, each with numerous problems, below. This is not meant as an exhaustive list requiring change but rather as representative of the quality of the writing found in the paper. I did not have to look hard to find examples -- these are the first sentence of the abstract and the last sentence of the first paragraph of the introduction, respectively.

> Recently, multi-agent reinforcement learning (MARL) adopts the centralized training with decentralized execution (CTDE) framework that trains agents using the data from all agents at a centralized server while each agent takes an action from its observation.

> Hence, the multi-agent reinforcement learning (MARL) that operates in multi-agent domain has been introduced and now becomes one of the most active and challenging RL research area.

### Centralized vs Decentralized MARL
The submission does not accurately characterize the disadvantages of decentralized training (see below). Using a centralized value function during training neither resolves non-stationarity nor partial observability.

> In MARL, the decentralized approach has been used to train each agent based on its trajectory (Tan, 1993). However, it often shows unstable and low performance due to non-stationary environment and partially observable information (Tan, 1993; Foerster et al., 2017) that inherits from the decentralization. Specifically, as the agents evolve their policies independently, the environment becomes non-stationary, which unstabilizes training at each agent. In addition, the agent may not observe the information of other agents, which causes low performance in the cooperative or competitive environment

Contrary to the tone of the submission, centralized value functions are not strictly beneficial. See Contrasting Centralized and Decentralized Critics in Multi-Agent Reinforcement Learning (2021).

### Problem Setting
I don't find the problem setting suggested by the submission to be well motivated. Could the submission give an application setting  that is currently addressed by decentralized MARL algorithms in which centralized MARL algorithms are inapplicable because of missing data? I think having such an example would help add credibility to the proposed problem setting.

### Proposed Methodology
The proposed methodology is fairly simple (simply combining an imputation method with some masking). However, I do not think that that is a problem, so long as the problem setting is aptly justified.

### Experiments
The submission shows some evidence that IA-MARL can outperform independent DDPG. However, for IA-MARL to merit the implementation trouble, it needs to offer better performance than a host of (non-centralized) MARL algorithms (e.g., independent PPO, independent R2D2 (minus the distributed part)), not just DDPG. I also concur with reviewer SVEY regarding the importance of of the baseline that inputs an "unobserved token" for unobserved data.

The submission states that it does not perform experiments on SMAC because MADDPG does not perform well in SMAC. I find this reasoning somewhat lacking. If IA-MARL is MADDPG-specific (which it isn't) and MADDPG does not perform well in some settings, that would be a significant downside for IA-MARL. But given that IA-MARL is not MADDPG-specific, and (as also noted by reviewer SVEY), MAPPO performs well on SMAC, its not clear why SMAC isn't an appropriate testbed.

Hyperparameter tuning can make a big difference in the performance of MARL algorithms. The appendix has some information about what hyperparameters the submission used but appears to be lacking information about how each algorithm examined in the submission was tuned. This information is especially important (in my opinion at least) because the submission is examining a novel problem setting in which there do not exist externally established performance metrics against which to compare.

### Question

It is common practice in the cooperative MARL community to share parameters between agents (even when the agents are not using a centralized value function). Is this allowed in the problem setting proposed by the submission?

**Summary Of The Paper:**

The submission proposes a cooperative MARL problem settings in which the observation-action-reward tuples generated during training are unavailable with some non-zero probability. The submission suggests addressing this problem setting by first imputing the missing training data and what they call a mask based update. The submission presents experiments for multi-agent particle environments.

**Summary Of The Review:**

Overall, I think the submission would need to

- Do more to justify the idea that this is a problem setting in need of attention
- Address the deficiencies in writing quality
- Add additional relevant baselines / experiments
- Add additional details regarding hyperparameter tuning

to merit acceptance.

---

> ### Author Response · Authors · 2021-11-23
> **Response to Reviewer vfgR**
>
> - Centralized vs decentralized critics
>
> We have taken a close look at the paper Lyu et al. (2021). Although we agree with the reviewer that the centralized value functions might not be strictly beneficial, one cannot also firmly conclude that the decentralized critic always outperforms the centralized critic as also claimed in Lyu et al. (2021). It certainly depends on the environment settings and algorithms and we have provided the experimental results that the centralized critic performs better than the decentralized critic in our setting by showing both MADDPG (a centralized critic) and IA-MARL (our proposed approach) outperformed DDPG (a decentralized critic).
>
>
> - Problem settings and proposed methodology
>
> This paper tackles the missing training data problem for CTDE-based MARL, not for the decentralized critic-based MARL. In the decentralized critic-based MARL, the issue does not exist because agents do not have to transmit their data to the central server. Nevertheless, we also have shown that DDPG (decentralized critic) performs worse than MADDPG (centralized critic) even with the missing training data.
>
> - Experiments
>
> 1) Unobserved token - As the reviewer points out, unobserved token can be used for the missing training data. We newly added the corresponding results in the Appendix B, where IA-MARL and Simple-IA-MARL with the unobserved token show lower performance than those using the GAIN.
> 2) Hyperparameter – We fine-tuned the learning rate and the number of hidden nodes for the DDPG and MADDPG in the Speaker-Listener environment through the grid search and use the hyperparameters for both the Speaker-Listener and the Tag environment. Since our focus is to show the effect of missing training data and IA-MARL for a given algorithm, i.e., MADDPG, we show the results with different missing training data, hyperparameters for the imputation, and ablation study.
> 3) PPO and Other MARL algorithms – Due to the limited time and resources, we were not able to add the experiment results of independent PPO and other MARL algorithm for SMAC environment in this paper. However, we believe that the performance of IA-MARL has been well verified from the current results of DDPG and MADDPG.
>
> - Question
> The proposed problem of the missing training data can happen in the cooperative MARL when agents share the parameters with each other. In the missing training data problem, we get the gradients of the shared parameters $\phi_s$ as in (14). We can use the gradients of the shared parameter as $\frac{1}{n_\text{share}}\nabla_{\phi_s}J(\phi_s)$, where $n_{share}$ is the number of agents that share the parameters.

---

> > ### Comment · Reviewer_vfgR · 2021-11-24
> > **Follow Up**
> >
> > I have read the author response and looked at the revised version of the submission.
> >
> > ### Writing
> >
> > The submission does not appear to have made any changes to its writing. Even the two sentences explicitly given above remain uncorrected.
> >
> > > We have taken a close look at the paper Lyu et al. (2021). Although we agree with the reviewer that the centralized value functions might not be strictly beneficial, one cannot also firmly conclude that the decentralized critic always outperforms the centralized critic as also claimed in Lyu et al. (2021). It certainly depends on the environment settings and algorithms and we have provided the experimental results that the centralized critic performs better than the decentralized critic in our setting by showing both MADDPG (a centralized critic) and IA-MARL (our proposed approach) outperformed DDPG (a decentralized critic).
> >
> > I agree that "one cannot also firmly conclude that the decentralized critic always outperforms the centralized critic." However, the reason I made this comment was in response to the fact that the submission reads as if the authors believe that centralized value functions are strictly superior to decentralized value functions. Indeed, (even in the revised version!) the submission makes statements such as "Recently, the centralized training with decentralized execution (CTDE) framework has been introduced for MARL (Oliehoek et al., 2008; Foerster et al., 2018). This can alleviate the non-stationary environment and the partially observable information problems." I would characterize this claim as misleading, if not untrue.
> >
> > ### Problem Setting
> >
> > The submission writes the following regarding my comments on problem setting.
> >
> > > This paper tackles the missing training data problem for CTDE-based MARL, not for the decentralized critic-based MARL. In the decentralized critic-based MARL, the issue does not exist because agents do not have to transmit their data to the central server. Nevertheless, we also have shown that DDPG (decentralized critic) performs worse than MADDPG (centralized critic) even with the missing training data.
> >
> > I don't disagree with most of this, though I'd say the submission's methodology is specifically for centralized value functions, rather than general CTDE-based MARL algorithms (eg, it would not work for BAD or CAPI).
> >
> > Perhaps there was a misunderstanding regarding my point above. I asked for an example of a setting in which practitioners are forced to us decentralized MARL because, as a result of the missing data problem, CTDE cannot be employed.
> >
> > ### Experiments
> >
> > > Unobserved token - As the reviewer points out, unobserved token can be used for the missing training data. We newly added the corresponding results in the Appendix B
> >
> > Thanks for adding this experiment.
> >
> > > we were not able to add the experiment results of independent PPO ... we believe that the performance of IA-MARL has been well verified from the current results of DDPG and MADDPG.
> >
> > I respectfully disagree here. If IA-MARL cannot outperform strong independent learning baselines, it does not justify the added complexity (at least in my opinion).
> >
> > ---
> > In response the revisions and author comments above, my score remains unchanged. I don't feel that my concerns regarding the problem setting, the writing, or the experiments were addressed to the extent that I was hoping to see.

---

### Official Review · Reviewer_SVEY · 2021-10-30

**Correctness:** 3
**Technical Novelty And Significance:** 2
**Empirical Novelty And Significance:** 3
**Recommendation:** 3
**Confidence:** 4

**Main Review:**

Strengths:
- Simple, intuitive method that seems fairly effective in simple toy settings.

Weaknesses:
- Experimental results:
1) Missing ablations and baselines: For every graph I would expect a comparison to state-of-the-art decentralized methods, such as independent PPO that only train on those transitions that are available for a given agent (i.e. not missing data). There is also a missing trivial baseline which simply inputs a "not observed token" (e.g. -1) for unobserved data and otherwise does standard CTDE. The current "random imputation baseline" samples from the entire range which seems like a strawman.
2) No large scale experiments: The authors argue that SMAC can't be used as a testbed since MAPPO does poorly there. However, they also state that "other actor-critic RL methods including policy gradient are also applicable for IA-MARL". Multi-agent PPO is known to do well on SMAC, so why not test there? It's also unclear why the method should only be restricted actor-critic variants. What prevents the method from being used with e.g. VDM or QMIX?
3) Clarity: It would be great to explain what the shading is in the plot. I assume this is standard error of the mean? The results also seem really noisy and somewhat hard to interpret.  "we use DDPG for the prey and MADDPG for predators": This statement is confusing. The plots for the tag environment contain MADDPG lines for both the prey and predators. Is this supposed to mean that each method is evaluated _against_ MADDPG predators and DDPG preys?

- Conceptual:
The fact that masking is needed to make the method work is a red flag. It clearly illustrates that the imputation of the data does not actually recover the correct values. In principle, under centralized training, having wrong values for _the other_ agents can also be detrimental to learning, which would render the current masking ineffective. The fact that this doesn't happen in these problem settings seems like an accident and it would be great to explain any assumptions that make this possible.
A good workaround for all of this might be to simply feed the imputation mask along with the confidence of the discriminator to both the agents and the value function. That way they can learn to utilise the data appropriately.


**Summary Of The Paper:**

This paper develops an imputation method for missing data in multi-agent settings. The main idea is to train a GAN based generator that imputes missing data in the the centralised learning with decentralized execution (CTDE) setting.

**Summary Of The Review:**

The method currently seems fairly incremental and it's unclear to me how broadly applicable it is. This is both due to the limited /noisy experimental evaluation and the missing discussion regarding the assumptions that went into the method and limitations.

---

> ### Author Response · Authors · 2021-11-23
> **Response to Reviewer SVEY**
>
> 1.
>
> As the reviewer pointed out, the unobserved token can also be used for the missing training data. We newly added the results of this case in the Appendix B, which shows that IA-MARL and Simple-IA-MARL with the unobserved token performed worse than those using the GAIN.
>
> 2.
>
> We specify that “other actor-critic RL methods including policy gradient are also applicable for IA-MARL” since we can easily apply the mask-based update for the actor-critic RL method using equations (13) and (14) but some modifications are also required for other algorithms. The techniques used in IA-MARL (i.e., the imputation and the mask-based update) can be used for other MARL algorithms.
> When the IA-MARL is applied for VDN and QMIX, we should modify the mask-based update as VDN and QMIX do not have policy. For instance, in VDN, the utility of agent $i$ is $V_i(o_{i,t}, a_{i,t})$, where the sum of utility is trained to minimize the return, i.e., loss of $\theta_i$ $L_{\theta_i}=(R_t-\sum_{i=0}^nV_i(o_{i,t}, a_{i,t}))^2$. To train the agent $i$ using the mask-based update, we can use $L_{\theta_i}$ when $(o_{i,t}, a_{i,t}, r_t, o_{i,t+1}, a_{i,t+1})$ exists.
> We have clarified this point in the revised paper.
> Due to the limited time and resources, we were not able to add the experiment results of independent PPO and other MARL algorithms in this paper. However, we believe that the performance of IA-MARL has been well verified from the current results of DDPG and MADDPG, and this work has a contribution for the environments where the decentralized critic-based MARL are used.
>
> 3.
>
> The shading in plots is a confident interval, and we have clarified it in the revised paper. We have also replaced "we use DDPG for the prey and MADDPG for predators" with “In tag environment, we evaluate IA-MARL against MADDPG predators and DDPG prey” to avoid the confusion.
>
> 4. Conceptual suggestion
>
> We agree with the reviewer that having wrong data for other agents (i.e., the imputed data with low accuracy) is detrimental to learning. This has also been shown in our experiment results in Figs. 2 and 3, which show the average rewards for different numbers of pre-train in GAIN, i.e., $N_{pre}$. Lower $N_{pre}$ results in lower accuracy in the imputed data, and in this case, IA-MARL did not perform well and we discussed about the required pre-trained number of GAIN in Section 5.1.
> However, if the mask-based update is not used, the performance of IA-MARL can be worse as shown in our ablation study in Section 5.3, even when the accuracy of the imputation is low. Therefore, we still think the mask-based update is effective and an important component in IA-MARL.
> As the reviewer mentioned, the output of the discriminator can also be used to improve the performance. To verify this, we conducted additional experiments. We trained and used the value function $V(\hat{o}_{t}, \hat{a}_t, \hat{M}_t)$, where $\hat{M}_t$ is the output of the discriminator. We found that this method has similar performance with the IA-MARL but there is no clear performance improvement.

---

### Official Review · Reviewer_oFfr · 2021-11-02

**Correctness:** 3
**Technical Novelty And Significance:** 2
**Empirical Novelty And Significance:** 2
**Recommendation:** 3
**Confidence:** 3

**Main Review:**

Strengths:
1. Clear writing in the Introduction. The intro clarifies some of the important issues that inspired in this paper, the motivation, possible solutions, etc.
2. The problem of MARL under missing data is interesting.
3. The experiments are conducted under widely-used MARL benchmark environments.


Weaknesses:
1. Unconvincing experiments. This paper has abundant experiments, from different percentages of missing data to different environmental settings. However, I'm curious to see how the MADDPG without imputation works against with imputation. Maybe the  MADDPG without imputation itself could work well enough, and there is no need to impute. Since the missing data are randomly set, there are going to be cases where there is no missing data across all agents at that time. The authors should add this experiment, also with different missing rates, to convince the readers that imputation does work.

2. The generality of the proposed method to a bigger domain. The imputation method used by this paper is GAIN, which is originally focused on vectorized data. This paper proposed to use GAIN to impute the trajectory, which includes states, actions, and rewards. The relationships between states/actions/rewards themselves could be complicated enough to model, and I wonder how this imputation could handle the relationships. What's more, when the state input is images, like in some settings in SMAC, generating an image would be difficult. Therefore, I'm concerned with the proposed method could only be applied when the environment setting is too simple to match the arguments mentioned in the intro: the communication failure, hardware limit, and security attacks in in wireless sensoror other real-world applications.


**Summary Of The Paper:**

This paper proposes an imputation-assisted multi-agent reinforcement learning (IA-MARL) method under the problem setting of MARL, where the training data of each agent can be randomly missed with a certain probability. Specifically, IA-MARL uses a generative adversarial imputation network to impute the missing data, and train the MARL method on the imputed data.

**Summary Of The Review:**

Based on the weaknesses mentioned in this paper, I tend to rate this paper as "marginally below the acceptance threshold". The main concerns are located in the experimental results and the complexity of the imputation problem itself. But I'm open to change my score after reading the rebuttal from the authors.

---

> ### Author Response · Authors · 2021-11-23
> **Response to Reviewer oFfr**
>
> 1.
>
> We agree to the suggestion of the reviewer. In the paper, we actually provided the experiment results of MADDPG without imputation in the ablation study (Section 5.3) and Figure 4 (please see the results of IA-MARL (random)). Even with the random filling (i.e., random imputation), IA-MARL works better than the Simple-IA-MARL that does not use the mask-based update. However, it performs worse than IA-MARL with imputation.
> In addition, if neither the imputation nor the random filling is used, the sample efficiency will decrease significantly as we show the number of training data in equation (17) and the learning curves in Appendix B. When the training data is missing, the number of training data for MADDPG decreases exponentially with the number of agents, therefore, the training would be much slower than IA-MARL.
>
> 2.
>
> The imputation methods replace the missing data with an estimated value based on other available information. For instance, in the Tag environment, the position of agent $i$ at time $t$ can be imputed with the position at time $t-1$ and $t+1$, where the GAIN is trained to minimize the difference between the missed data and the imputed data.
> Furthermore, we do not think that GAIN (or modern generative models) is not capable of imputing the image observation or other states in the environments. We have seen great success in image generations with the learning-based methods and also recent model-based RL algorithms have shown that we can model complex environments, e.g. a model that predicts the image of the next time step and the reward in Lukasz et al (2020).
> Lukasz et al (2020): Lukasz Kaiser et al. Model-based reinforcement learning for Atari. In International Conference on Representation Learning, 2020.

---

### Official Review · Reviewer_XnkH · 2021-11-08

**Correctness:** 3
**Technical Novelty And Significance:** 4
**Empirical Novelty And Significance:** 3
**Recommendation:** 5
**Confidence:** 4

**Main Review:**

The strength of the IA-MARL paper lies in its problem specificity and the natural alignment between agent multiplicity in MARL research and the motivation for imputation: filling in the blanks. The flow and structure make the paper easy to read, and the imputation approach appears convincing and effective and poses some intriguing discussion points, which in my view could also constitute the paper's weaknesses. I would appreciate if the authors could include some of these discussion points in the paper, where they see fit.

Q1. What would you say is the strongest motivation for employing imputation techniques in the multi-agent case? Do missing training data problems occur more frequently or cause more severe damage in the multi-agent case?

Q2. In the single-agent case, if the imputation could be applied across the time axis (and not rely on the observations of other agents because there would be none), what benefits would you foresee? As a follow-up, how much would you say does IA-MARL take advantage of the information across the time axis or across the agent axis? In other words, when filling in the blanks for agent i, does IA-MARL learn more from the observation history of that agent, or from the current observations of other agents?

Q3. As the authors imply, it is becoming more important to take into account real-world limitations when deploying MARL (e.g., communication failure). Could you discuss some relevance to a 2019 ICLR paper called SchedNet by Kim et al.? (which, to the best of my knowledge, is one of the earliest MARL works addressing real-life communications constraints.

Q4. If time allows for it, it would be valuable to observe how IA-MARL stacks up against some of the more recent MARL methods of the authors' choice. I understand that none of them were designed to tackle the missing training data problem, but it may still be worthwhile to assess how they compete against each other in the missing training data setting: some works that come to my mind include: Deep Coordination Graphs, MAVEN, Bayesian Action Decoders, Stable Opponent Shaping, and QPLEX.

Q5. I think the presented solution works well, but I am not entirely persuaded about how severe the missing training data problem is, possibly because I am not a security person. However, I do think that certain additions might strengthen the problem and the motivation by a large extent. For instance, Q5-a wouldn't you say that a poorly exploring team of agents is encountering a missing training data problem? (in the sense that all agents' training data is missing after time t). This case ties in with Q2; if IA-MARL learns across the time axis, then it could also carry out imputation in future time indices? Now, Q5-b, how would IA-MARL perform in situations where agent homogeneity does not hold? In the weak heterogeneity case, agents may have the same observation horizon (e.g., 3x3 grid around them) but some of them might have higher chance of missing data (e.g., due to poor comms link quality). In the strong heterogeneity case, agents could have different missing data ratios as well as different observation horizon, different observation frequency (e.g., some report at 1Hz and some others at 2Hz), and/or all of the aforementioned properties could vary across time (e.g., as comms link quality may also vary in time in e.g., FANETs). How would the authors find these kinds of factors with regard to the missing training data problem? (e.g., are they relevant/motivating or irrelevant?) I think these cases could help strengthen the missing training data problem the authors are trying to build; I would love to hear from the authors.

**Summary Of The Paper:**

Authors present a novel and effective method leveraging a generative adversarial approach to a specific MARL problem with regard to missing training data. The approach presented treats the missing data as targets of imputation, loosely similar to that of inpainting problems in computer vision, where missing pixels are "filled in". Results show that IA-MARL outperforms one baseline (MADDPG) that has not been originally devised for missing training data.

**Summary Of The Review:**

IA-MARL solution works well, but the missing training data problem does not look scary, as presented in the paper. I would gladly buy almost all of the claims presented, except the motivation behind the problem itself. To be specific, I will up my score if I could learn more from the authors about Q5 and at least one from Q1~Q4.

---

> ### Author Response · Authors · 2021-11-23
> **Response to Reviewer XnkH**
>
> 1.
>
> The main motivation of this work is that the missing training data problem causes severe damage in the centralized training with decentralized execution (CTDE) framework. Specifically, when we use MADDPG and train the value function in (3), the data from all agents should exist. Hence, the missing training data of certain agents makes the obtained data from other agents unusable and this could degrade the sample efficiency significantly.
> This missing training data problem can also happen for single-agent cases or decentralized training cases. In the single-agent cases, the number of obtained data is $N_\mathcal{D}$ and the number of training data is $N_\mathcal{D}(1-p_m)^{2}$, where $p_m$ is the missing probability. However, when the number of agents is $n>1$, the number of training data is $N_\mathcal{D}(1-p_m)^{2n}$, which exponentially decreases as the number of agents $n$ increases (please refer to equation (17) and Appendix B of our paper for more details). This shows the missing training data in CTDE framework is more serious than single-agent cases or multi-agent cases with decentralized training.
>
> 2.
>
> In IA-MARL, both the information of the time axis and the agent axis are used for the imputation (as in equation (7)). However, it is difficult to say which information benefits more and it depends on the applications of multi-agent systems and imputation algorithms.
> The imputation accuracy generally affects the performance of the IA-MARL, which we already have shown and discussed in Figures 2 and 3.
>
> 3.
>
> In Kim et al (2019), the authors proposed SchedNet for the environment having limited communication channels between agents “during execution”, but assumed that the training data always exists during training. Note that the communication between the agents during the execution is to encourage coordination among the agents. On the other hand, in our work, the training data collected from agents can be missed, which affects “training of agents”. To summarize, there are two differences between SchedNet and IA-MARL: 1) SchedNet considers limited communication during execution and IA-MARL considers data missing for the training 2) the number of messages in ScheNet is limited while IA-MARL cannot obtain the missed data. We will cite and discuss it in the related works.
>
> 5-a.
>
> We are afraid that we did not fully understand the comment. We guess the reviewer's intention was whether the imputation and IA-MARL work well even when the training data from all agents are missed. When the data from all agents are missed, the imputation for the future time indices is still possible, but imputation might not be accurate. Furthermore, in IA-MARL, we cannot train the agent when the data from all agents are missed. We use the mask-based update that trains the agent $i$ “only when the consecutive data of that agent is not missed”. In Simple-IA-MARL, agents can be trained with imputed data without the mask-based update, however, it shows low performance as shown in Section 5.3.
> (We did not claim a poorly exploring team of agents encounter the missing training data problem. The missing training data problem arises when the training data collection is not always perfect.)
>
> 5-b
>
> IA-MARL also works when agents have different observation spaces and different missing training data probabilities. In MPE environment, the agents may have different observation and action spaces. For instance, in the speaker-listener environment, the observation of the speaker is the target color while that of the listener is a message from the speaker. Furthermore, IA-MARL is designed to cover the training data missing problem even when the missing probability is different across the agent as specified in Section 4.1.
> When we need to consider different observation frequency and different environment properties, the IA-MARL does still works although reformulated problem and different algorithms should be used. For instance, when observation of agent $j$ is updated every 3 steps, we may set $o_{j,0}=o_{j,1}=o_{j,2}$. Also, given environmental properties like limited channels, we can use SchedNet in Kim et al (2019) instead of MADDPG for IA-MARL. Note that we use imputation for the missing data and mask-based update to further stabilize the training, and other actor-critic based algorithms can also be used.

---

### Decision · Program_Chairs · 2022-01-20

**Decision:**

Reject

**Comment:**

The paper tackles the problem of missing data in centralized training multi-agent RL approaches. The authors propose 1) using generative adversarial imputation networks for imputing missing data and 2) discarding training data where data from multiple consecutive timesteps is missing.

Reviewers agreed that the problem of missing data in multi-agent RL is interesting. At the same time, several reviewers shared two main concerns about the experimental evaluation:
* The lack of comparisons to baselines other than MADDPG, especially decentralized critic approaches.
* The lack of experiments on non-toy domains such as SMAC.

The author response did not sufficiently address these concerns leaving the reviewers in agreement that the paper should not be accepted without these additional experiments.